# The Role of KRAS Mutations in Colorectal Cancer: Biological Insights, Clinical Implications, and Future Therapeutic Perspectives

**DOI:** 10.3390/cancers17030428

**Published:** 2025-01-27

**Authors:** Mitsunobu Takeda, Shoma Yoshida, Takuya Inoue, Yuki Sekido, Tsuyoshi Hata, Atsushi Hamabe, Takayuki Ogino, Norikatsu Miyoshi, Mamoru Uemura, Hirofumi Yamamoto, Yuichiro Doki, Hidetoshi Eguchi

**Affiliations:** Department of Gastroenterological Surgery, Graduate School of Medicine, Osaka University, Osaka 565-0871, Japan

**Keywords:** KRAS, colorectal cancer, KRAS mutations, KRAS^G12C^ inhibitor, primary and acquired resistance, liquid biopsy

## Abstract

KRAS mutations are key drivers of colorectal cancer progression and resistance to treatment, significantly limiting therapeutic options for affected patients. Found in 30–40% of cases, these mutations promote persistent activation of oncogenic pathways such as MAPK/ERK and PI3K/AKT, contributing to tumor growth, poor prognosis, and reduced responsiveness to anti-EGFR therapies. This study aims to elucidate the biological role, clinical significance, and therapeutic potential of targeting KRAS mutations. Recent breakthroughs include the development of targeted inhibitors, such as sotorasib and adagrasib for G12C mutations, and experimental therapies for G12D. However, therapeutic responses in colorectal cancer remain suboptimal compared to other malignancies, largely due to resistance mechanisms and tumor heterogeneity. Promising strategies, including combination therapies, vaccines, and nucleic acid-based treatments, offer hope for improved outcomes. These findings underscore the importance of advancing personalized approaches to enhance care for patients with KRAS-mutant colorectal cancer.

## 1. Introduction

Colorectal cancer (CRC) afflicts over 2 million people worldwide every year. It represents the second leading cause of cancer deaths and ranks third in terms of incidence [1]. Colorectal cancer remains a leading cause of cancer-related mortality worldwide despite advances in early detection and treatment. Although the median overall survival (OS) for advanced recurrent colorectal cancer has exceeded 30 months due to advances in drug therapy and surgical techniques, some patients are still refractory to treatment and have a poor prognosis. As such, it is essential to further pursue biological and molecular investigations of colorectal cancer and ensure that these findings can be applied in clinical practice [2]. The characteristic genetic alterations in CRC include activating mutations in the oncogene KRAS (42%) and inactivating mutations or deletions in the tumor suppressor genes APC (75%) and TP53 (60%) [3]. However, among the numerous genetic alterations associated with CRC, mutations in the KRAS gene are particularly significant.

The oncogene KRAS has been a subject of research in human malignancies since its discovery in the 1960s [4,5]. KRAS mutations are observed in over 20% of human cancers, though the frequency of these mutations varies across different ethnic groups and different anatomical sites of CRC [5].

KRAS, a GTPase within the RAS family, plays a pivotal role in regulating various cellular processes such as growth, differentiation, and survival, as outlined in the following sections. Mutations in KRAS, particularly those occurring at codons 12 and 13, lead to the persistent activation of the KRAS protein, thereby driving tumorigenic signaling pathways. This article aims to offer an in-depth review of the significance of KRAS mutations in colorectal cancer, exploring their biological implications, clinical relevance, and potential as therapeutic targets.

## 2. Biological Significance of KRAS Mutations

In humans, there are four isoforms of the RAS protein: HRAS, NRAS, and two splice variants, KRAS4A and KRAS4B [6]. Among these, KRAS is the most frequently mutated RAS gene in CRC [7].

KRAS is a member of the guanine nucleotide-binding protein family [4] and cycles between an inactive guanosine diphosphate (GDP)-bound state (“off”) and an active guanosine triphosphate (GTP)-bound state (“on”). In its GDP-bound form, KRAS is inactive. When a receptor is activated, guanine nucleotide exchange factors (GEFs) are stimulated, promoting the exchange of GDP for GTP. In the GTP-bound state, KRAS activates downstream effectors, initiating multiple signaling pathways, including the RAF kinase family and phosphatidylinositol 3-kinase (PI3K), which promote various cellular functions [8,9] (Figure 1).

Under normal conditions, KRAS activation is a transient process due to the activity of the intrinsic enzyme GTPase, which inactivates the pathway by converting RAS-GTP to the inactive form RAS-GDP. However, most somatic mutations in KRAS impair this GTPase function, leading to a persistent increase in RAS-GTP levels. This impairment results in the continuous activation of RAS effectors and the PI3K signaling pathway, ultimately driving uncontrolled cell proliferation—a hallmark of malignancy [10,11].

The structure of the GTP/GDP binding site in KRAS is crucial for maintaining proper regulatory control. Even a single amino acid substitution at this site can disrupt normal regulatory mechanisms. Notably, point mutations at codons 12 and 13 are common and typically affect the glycine residues within the GTP-binding pocket, which play a critical role in GTPase activity. These mutations stabilize KRAS in its active state for prolonged periods, amplifying downstream signaling pathways [2].

KRAS mutations in colorectal cancer lead to the sustained activation of critical downstream signaling pathways, including the MAPK/ERK and PI3K/AKT pathways. These pathways promote uncontrolled cell proliferation, inhibit apoptosis, and support tumor growth and metastasis [12]. Overactivated KRAS proteins interact with various effector proteins, driving persistent activation of these signaling cascades. Furthermore, KRAS mutations contribute to tumor initiation and progression [13].

Structurally, KRAS belongs to the small GTPase family and contains a highly conserved N-terminal G-domain of approximately 170 residues and a C-terminal hypervariable region (HVR) of approximately 25 residues. This G-domain is composed of five alpha-helices and six beta-strands, and forms a catalytic core that is responsible for GTP binding and hydrolysis [14,15]. There are several important regions in this domain.

P-loop (phosphate-binding loop): This loop (residues 10–16) stabilizes the phosphate group of GTP or GDP.

Switch I (residues 32–38): This region undergoes a conformational change during GTP hydrolysis and is essential for interaction with effectors such as RAF.

Switch II (residues 59–67): This region adopts a different conformation in the GDP-bound and GTP-bound states. Although this region has been the focus of much drug discovery research in recent years, it is not the only region important for KRAS function.

Additional loops and regions: Beyond switch I and II, other structural elements regulate KRAS’s affinity for nucleotides and its ability to bind various regulators and effectors (e.g., guanine nucleotide exchange factors, GEFs, and GTPase activating proteins, GAPs). Genetic analysis has shown that 98% of RAS mutations that cause cancer are concentrated in three residues: G12, G13, and Q61. All of these are located near the phosphate group of GTP (p-loop: G12 and G13, Switch II: Q61), and it is thought that mutations in these residues stabilize the GTP-bound form and cause constant signal activation.

The most frequently detected mutations in codons 12 and 13—namely G12V (incidence: 20–25%), G12S (5–7%), G12R (3–5%), G12A (3–5%), and G13D (15–20%)—are also clinically relevant, with their prevalence varying by tumor type [16,17,18]. While G12C and G12D have garnered significant attention due to the development of novel small molecule inhibitors [19,20], other substitutions remain relatively underexplored in terms of direct targeted therapies.

For example, G12V is frequently identified in pancreatic ductal adenocarcinomas and is associated with aggressive disease biology [16,21]. G12R, which is also common in pancreatic cancer, displays altered biochemical properties—particularly in GDP/GTP hydrolysis—suggesting that unique conformational features may influence inhibitor design [22]. Moreover, G12S and G12A occur in both colorectal and lung cancers at lower but still significant frequencies [16,17]. These variants often retain high GTP affinity and can activate downstream effectors in a manner that is subtly distinct from G12C or G12D [23]. Finally, G13D—while less frequently discussed—represents a non-negligible subset of RAS mutations, especially in colorectal cancer, where it may respond differently to anti-EGFR therapies compared with other mutations [24].

## 3. Emerging Roles of KRAS Mutations in Colorectal Cancer: Implications as a Therapeutic Target

Recent studies have uncovered the various roles of KRAS in colorectal cancer, highlighting its potential as a new therapeutic target. Below are four recently reported functions of KRAS mutations.

### 3.1. KRAS Influences the Tumor Microenvironment

In the tumor microenvironment, cancer-associated fibroblasts (CAFs) have garnered attention for their role in promoting cancer progression. CAFs support the transformation of normal epithelial cells into cancerous cells and facilitate the migration and metastasis of cancer cells by conferring a stromal-like phenotype [25]. KRAS can alter the phenotype of CAFs, enriching them with lipids, which promotes angiogenesis and tumor progression. KRAS activates the transcription factor CP2 (TFCP2), which then stimulates adipocyte differentiation factors, converting CAFs into lipid-rich cells. These lipid-enriched CAFs produce VEGFA, fostering angiogenesis and supporting tumor growth [26].

### 3.2. KRAS Promotes Cancer Stem Cell Formation

The presence of cancer stem cells (CSCs) has been suggested in solid tumors, similar to hematologic malignancies, making them a focal point as potential targets for cancer eradication. CSCs contribute to the hierarchical structure of cancer tissue and play a crucial role in tumor heterogeneity and the acquisition of treatment resistance [27]. Recent studies have shown that oncogenic RAS mutations induce cancer stem cell formation by activating CDK1 and promoting O-GlcNAcylation of proteins, which upregulates SOX2 expression, a key factor in CSC generation [28].

### 3.3. KRAS Activates Macropinocytosis (MP)

Macropinocytosis (MP) is a type of endocytosis that enables cells to acquire extracellular nutrients essential for survival. This process involves the internalization of serum proteins and necrotic cells, serving as a critical energy source, particularly for cancer cells [29]. MP also plays a role in chemoresistance [30], with certain cancer cells relying on MP for nutrient acquisition. Recent findings have shown that RAS mutations and the PI3K pathway contribute to MP activation. In RAS-mutant colorectal cancer, MP is indeed upregulated, and these cancer cells exhibit sensitivity to MP inhibitors [28,30].

### 3.4. KRAS Involvement in Cell Competition in Colorectal Cancer

Cell competition is defined as the elimination of certain cells through interactions between two distinct cell populations in close proximity. Originally observed as the removal of mutant cells surrounded by wild-type cells, some mutant cells have been found to act as “super-competitors”, eliminating neighboring wild-type cells [31]. In colorectal cancer, where Wnt and Ras signaling are frequently activated, this process of cell competition is altered. In affected epithelial tissue, the production of diffusely invasive cancer cells is promoted. The activation of NF-κB signaling through innate immune receptors RIG-I/TLR3 enhances MMP21 production, enabling RAS-mutant cells to eliminate neighboring cells through cell competition and diffusely invade surrounding tissues [32].

In summary, KRAS plays multiple roles in colorectal cancer, making it an intriguing area for future research and exploration.

### 3.5. Rewiring of Cancer Metabolism by Oncogenic KRAS

Oncogenic KRAS significantly reprograms cellular metabolism, enhancing glycolysis, glutaminolysis, and aspartate metabolism to fuel tumor growth and survival. Recent findings have highlighted that KRAS mutations drive the upregulation of key enzymes, such as PCK1, PCK2, and ASS1, which are involved in aspartate and urea cycle metabolism, contributing to the metabolic flexibility of KRAS-mutant CRC cells [33,34,35]. These adaptations provide critical precursors for nucleotide biosynthesis and support redox homeostasis, making them attractive targets for therapeutic intervention.

## 4. Clinical Significance of KRAS Mutations (Role in Practice and Therapeutics)

The presence of KRAS mutations in colorectal cancer carries significant clinical implications. Approximately 30–40% of colorectal cancer cases harbor KRAS mutations, which are associated with a poor prognosis and resistance to certain treatments, including anti-EGFR monoclonal antibodies (panitumumab and cetuximab). Identifying the status of RAS mutations is essential for determining treatment strategies and predicting patient prognosis.

In clinical practice, genetic testing for mutations in genes, such as RAS and BRAF, is performed on biopsy or surgical samples prior to initiating chemotherapy, enabling personalized treatment regimens tailored to each patient. The routine assessment of RAS gene status before chemotherapy in clinical practice is underpinned by findings from numerous clinical trials. For unresectable colorectal cancer, the prognosis without treatment is typically around 8–12 months. However, with treatments such as FOLFOX or FOLFIRI, which combine 5-FU, oxaliplatin, and irinotecan, survival has been extended to approximately 20 months [36]. Moreover, the addition of molecular-targeted agents, such as bevacizumab, cetuximab, or panitumumab, has extended survival beyond 30 months [37,38,39]. By 2015, results from multiple clinical trials had revealed that bevacizumab is effective regardless of RAS gene status, while anti-EGFR antibodies are ineffective in RAS-mutant cases. In the past decade, it has also been shown that the biology and drug sensitivity of tumors differ between right- and left-sided colon cancers [40], and cases achieving Early Tumor Shrinkage (ETS) with anti-EGFR antibody therapy have shown favorable outcomes [41]. In light of these findings, the most effective first-line regimen for wild-type RAS, left-sided colorectal cancer is doublet chemotherapy (FOLFOX or FOLFIRI) combined with an anti-EGFR antibody. In contrast, for RAS-mutant or right-sided colorectal cancers, triplet therapy (FOLFOXIRI, XELOXIRI) plus bevacizumab, or doublet therapy (FOLFOX, FOLFIRI, CAPOX, SOX) plus bevacizumab, is recommended. Although survival has improved over time for wild-type RAS patients, those with RAS mutations continue to show poor response rates to targeted therapies, with their overall survival remaining at 15–17 months (Figure 2). Therefore, developing new therapeutic strategies to improve outcomes for RAS-mutant colorectal cancer is essential.

Furthermore, recent reports have linked RAS gene status not only to the efficacy of first-line anti-EGFR antibody therapies but also to response rates for third-line chemotherapy with trifluridine/tipiracil (FTD/TPI). This highlights the increasing role of RAS as a biomarker for predicting the effectiveness of drug therapies. In general, right-sided colorectal cancers exhibit both higher aggressiveness and lower treatment responsiveness than left-sided cancers, which may be related to the significantly higher incidence of RAS mutations in right-sided tumors compared to left-sided ones [42]. As such, RAS is closely linked with prognosis and drug resistance in colorectal cancer, and RAS mutations represent not merely unfavorable prognostic markers but rather a central focus in an evolving research process driven by an unmet clinical need.

KRAS mutations are also frequently observed in various other cancers and play an essential role in tumor survival, driving early efforts in therapeutic development. Despite the discovery of RAS mutations as oncogenic drivers nearly 40 years ago, the lack of effective inhibitors has made RAS mutations a prime example of “undruggable” targets. Due to issues such as the binding affinity between GTP and RAS proteins and the structure of the proteins, KRAS was undruggable for these periods. Specifically, this was due to several reasons. First, the development of RAS inhibitors requires disrupting the binding of GTP to RAS proteins, a binding affinity occurring at the picomolar level (10^−12^ M). In contrast, kinase inhibitors have traditionally succeeded by targeting ATP–kinase binding at the micromolar level (10^−6^ M), which involves a much lower specificity threshold than RAS inhibition. The second reason lies in the flat surface of RAS proteins, which lacks the binding pockets necessary for compound attachment. Third, agents targeting RAS effector signaling, such as MEK and PI3K inhibitors, which were developed due to the difficulty of directly inhibiting RAS, have not shown efficacy in clinical trials due to feedback mechanisms. These features collectively contributed to the long-held view that directly targeting KRAS was beyond reach.

Recently, however, breakthrough drugs targeting KRAS mutations, particularly the G12C mutation (e.g., sotorasib and adagrasib), have been developed. Overcoming the “undruggable” nature of RAS involved two key turning points. The first was the identification of compounds that bind irreversibly to KRAS^G12C^-GDP. Shokat’s group leveraged the high nucleophilicity and low abundance of cysteine residues in nature to conduct fragment screening against KRAS^G12C^-GDP, leading to the identification of a compound that binds covalently to cysteine, enabling KRAS^G12C^ to be trapped in an inactive GDP-bound state by occupying a novel pocket in the switch II region of the KRAS protein. The second turning point was the discovery of the histidine residue at position 95 (H95) at the top of the pocket. This enabled the optimization of compounds by targeting the groove formed by H95, enhancing the inhibitory effect to levels suitable for human applications.

Indeed, sotorasib demonstrated high efficacy in non-small cell lung cancer, with a response rate of 33.3%, stable disease rate of 57.9%, and disease control rate of 91.2% [43]. Adagrasib also showed promising results with a response rate of 42.9% [44]. Sotorasib received regulatory approval in the United States in 2021 and in Japan in 2022, while adagrasib was approved in the United States in December 2022. The development of KRAS^G12C^ inhibitors marks a significant breakthrough in treating RAS-mutant colorectal cancer [43,44]. However, the efficacy of RAS inhibitors remains low in colorectal cancer, with an overall response rate of only around 10% [44]. Furthermore, KRAS^G12C^ mutations are present in only 3–4% of colorectal cancer cases, leaving a limited pool of eligible patients for these treatments.

## 5. Challenges and Limitations of Current Treatment

### 5.1. Limitations of RAS Mutation Testing

As previously mentioned, in cases of unresectable, advanced, and recurrent colorectal cancer, it has been reported that if there is a mutation in any of the following codons of the RAS (KRAS/NRAS) gene—exon 2 (codons 12, 13), exon 3 (codons 59, 61), or exon 4 (codons 117, 146)—suppressing upstream stimuli such as EGFR cannot inhibit RAS activity. Therefore, treatment efficacy with anti-EGFR monoclonal antibodies, such as cetuximab and panitumumab, is not expected. Consequently, prior to the initiation of chemotherapy for unresectable, advanced, and recurrent colorectal cancer, confirming RAS gene mutation status to determine the suitability of cetuximab and panitumumab based on the presence or absence of RAS mutations is considered to be of high clinical utility, and RAS gene mutation testing is widely performed.

Point mutations in the RAS gene have been reported to occur early in colorectal cancer and are detected at a certain frequency regardless of the cancer stage (Table 1) [45,46]. The frequency of KRAS exon 2 (codons 12, 13) mutations is approximately 35–40% in colorectal cancer, with no differences observed between reports from Western countries and Japan (Table 1). According to clinical trials conducted primarily in Western countries, the combined frequency of mutations in KRAS exons 3 and 4, and NRAS exons 2, 3, and 4 is 10–15% (approximately 20% of KRAS exon 2 wild-type cases). Notably, mutations in NRAS exon 4 (codons 117 and 146) are extremely rare, occurring in less than 0.3% of all colorectal cancers, with no significant differences observed between Japan and Western countries (Table 2). Regarding rectal cancer, KRAS mutations are detected in approximately 35–45% of patients with locally advanced rectal cancer, with no significant differences in KRAS status between colorectal and rectal cancers [45].

Recent findings have shown that not only somatic mutations in KRAS codons 12 and 13 but also other hotspot mutations in KRAS exons 3 and 4, and NRAS exons 2, 3, and 4, are associated with clinical resistance to anti-EGFR therapy [47]. Therefore, validating RAS wild-type status, including codons 12, 13, 59, 61, 117, and 146 in exons 2, 3, and 4, is essential for establishing treatment strategies for advanced colorectal cancer (CRC) [46,48].

Methods for detecting RAS mutations include direct sequencing (e.g., Sanger sequencing), polymerase chain reaction (PCR), and next-generation sequencing (NGS). The direct sequencing method amplifies the target gene region from tumor DNA to determine the nucleotide sequence directly. The PCR method uses allele-specific primers to amplify specific mutations, allowing the determination of the presence of the targeted mutation. Although direct sequencing can detect unknown mutations, its sensitivity is limited, requiring a tumor content of at least 10–25% (detectable when tumor cells with RAS mutations constitute at least 10–25% of the total cell population in a sample with normal cells). Compared to PCR, the sensitivity of direct sequencing is considerably lower. Thus, when performing RAS mutation testing via direct sequencing, it is essential to ensure a high proportion of tumor cells by marking the area of interest and manually dissecting the tumor tissue from the marked area.

RAS mutation testing methods used in clinical trials (Figure 3) include direct sequencing (PRIME trial, 20,050,181 trial, 20,020,408 trial, and PEAK trial), pyrosequencing (FIRE-3 trial), BEAMing (CALGB80405 trial), and Luminex (RASKET trial). The sensitivity of these methods ranged from 10 to 25% for direct sequencing (the least sensitive) to less than 1% for BEAMing (the most sensitive), with other methods generally having sensitivities of 1–10% [49,50]. Regardless of the testing method or sensitivity, the lack of efficacy of anti-EGFR monoclonal antibodies in RAS mutant cases has been consistently reproduced. Thus, while the optimal sensitivity is still unclear, testing methods with a sensitivity of 1–10% should be considered.

In routine clinical practice in Japan, the RASKET-B (Luminex) method is commonly used, and recently, many facilities are performing RAS mutation testing with preoperative endoscopic biopsy samples. This approach is based on the understanding that RAS mutations occur early in colorectal cancer development, so the mutation status of the tumor’s mucosal and invasive regions is generally consistent, with a concordance rate of over 97% for KRAS exon 2 mutations between surgical specimens and endoscopic biopsy samples in the same case [51].

However, current testing methods face two major challenges. The first challenge is the detection sensitivity of the test itself. Next-generation sequencing (NGS) is difficult to use in routine clinical practice due to cost-effectiveness issues, and the PCR method used in routine practice detects only a single mutation encoded by the molecular probe, with other mutations either being undetected or causing difficulty in result interpretation due to changes in qPCR efficiency. Although high-frequency RAS mutations can be detected by using multiple fluorescent dyes and probes, not all RAS mutations can be identified. Thus, there remains a possibility, albeit low, that a case diagnosed as wild-type RAS and treated with anti-EGFR monoclonal antibodies might actually be a RAS mutant case that does not respond to the therapy. The second challenge is the detection sensitivity related to the sample quality. Artifacts may arise in samples submitted for RAS mutation testing due to fixation or prolonged storage in paraffin blocks. Most of these artifacts are G > A and C > T transitions due to cytosine deamination. In old paraffin blocks, these artifacts can exceed 10% of the allele frequency, potentially leading to false-positive results. Additionally, for diagnostic testing using preoperative biopsies, a sufficient amount of tumor cell content is required, necessitating multiple biopsy samples.

Thus, the clinical application of RAS mutation test results cannot achieve 100% accuracy due to factors such as testing methods, sample quality, and tumor cell content, leaving the current challenges unresolved.

### 5.2. Challenges in the Interpretation of RAS Mutations

In recent years, RAS mutation testing has become routine in colorectal cancer, with numerous reports on KRAS clonality and comparisons between primary and metastatic sites. RAS mutations are considered driver mutations, and it is generally believed that subclonal events are relatively rare in solid tumors. However, heterogeneous KRAS mutations in primary lesions have been observed in 10–20% of cases [52], with a higher discordance rate of 27% in regional lymph node metastases [52]. Although discordance in KRAS mutation status is rare in distant organ metastases (less than 10%) [52,53], pulmonary metastases have shown a markedly higher discordance rate (approximately 30%) compared to other sites [54,55]. Furthermore, conversion from both wild-type to mutant and mutant to wild-type KRAS have been observed in metastatic lesions [56].

This indicates that some primary and metastatic tumors with mutant KRAS may be subclonal, and subclones (mutant allele frequency) could potentially confer resistance to targeted therapy [57]. In fact, an analysis of circulating tumor DNA (ctDNA) from patients with KRAS wild-type tumors who received anti-EGFR antibody therapy revealed that 38% of these cases exhibited mutant KRAS, suggesting the clonal selection of mutant cells within metastatic lesions [58]. Another study analyzing metastatic lesions from patients who were resistant to EGFR antibodies identified the frequent emergence of KRAS mutations (55%) and a wide range of mutant allele frequencies (0.04–17.3%), further confirming the clonal selection of KRAS mutant cells during targeted therapy [59,60].

Additionally, among patients with confirmed RAS mutations in pre-treatment tumor tissue samples, 19% (91/478) did not exhibit RAS mutations in their plasma at the time of treatment modification. Of these, 10.2% (49/478) showed no other somatic mutations. These findings suggest that approximately 10% of colorectal cancer patients who were previously deemed unlikely to benefit from treatments involving anti-EGFR antibodies due to RAS mutations may actually have the potential to benefit from such therapies [47]. The factors associated with changes in RAS mutation status (mutant to wild-type) included the absence of liver or lymph node metastases and the presence of low-frequency RAS mutations (mutations other than KRAS exon 2) [61].

Currently, there are no guidelines defining the threshold level of mutant RAS that impacts treatment efficacy. Furthermore, it remains unclear which types of mutant RAS have the highest clinical utility. In fact, it has been suggested that not all KRAS alleles confer resistance to anti-EGFR therapy. A previous retrospective analysis indicated that colorectal cancer (CRC) patients with the KRAS^G13D^ mutation showed efficacy with first-line chemotherapy combined with cetuximab [62,63]. Therefore, it cannot be conclusively stated that the presence of a RAS mutation renders anti-EGFR antibody therapy ineffective and unusable. A future approach may involve the mandatory assessment of RAS mutations via liquid biopsy prior to changing treatment regimens.

### 5.3. Resistance in RAS-Targeted Therapy

Due to the high adaptability and evolutionary capacity of cancer cells, resistance commonly emerges in gene-targeted therapies. Resistance can be broadly classified into two types: primary resistance and acquired resistance.

#### 5.3.1. Primary Resistance

Primary resistance refers to cases where gene-targeted therapy fails to show efficacy from the outset, reflecting an inherent property of the tumor itself. While the response rate to EGFR inhibitors in EGFR-mutant lung cancer exceeds 70%, the response rate of KRAS^G12C^ inhibitors in colorectal cancer is approximately 40%, which is not as high as that observed for molecular-targeted therapies against other driver gene alterations.

The relatively low efficacy of KRAS^G12C^ inhibitors compared to other molecular-targeted therapies can be attributed to both drug-related and tumor-related factors. On the drug side, the current KRAS^G12C^ inhibitors may exhibit insufficient inhibitory activity against mutant KRAS. For example, in the development of treatments for ALK-positive lung cancer, the first-generation drug crizotinib was initially approved, but the second-generation drug alectinib, with enhanced ALK inhibitory activity, demonstrated high efficacy. Similarly, it is possible that the first-in-class KRAS inhibitors, sotorasib and adagrasib, are not the best-in-class. However, numerous reports suggest that tumor-related factors, specifically the biological characteristics of KRAS-mutant tumors, play a significant role in the observed variations in efficacy. Here, we introduce three mechanisms of primary resistance.

Feedback Mechanism (Activation of Alternative Signaling Pathways)

When KRAS^G12C^ inhibitors suppress the MAPK signaling pathway, a feedback mechanism activates upstream receptors. These activated receptors can re-stimulate MAPK signaling through wild-type RAS (HRAS and NRAS), rendering tumor cells resistant to the inhibitor [64]. Such MAPK reactivation is observed within 24–72 h after inhibitor administration and is referred to as adaptive resistance. Furthermore, it has been reported that newly synthesized KRASG12C protein post-inhibitor treatment can be activated by feedback mechanisms involving EGFR or Aurora kinase, shifting to a GTP-bound form and thus bypassing the GDP trap induced by KRAS^G12C^ inhibitors [43]. These feedback mechanisms are activated shortly after administration, contributing to primary resistance.

2.KRAS-Mutant Tumors’ KRAS Independence

Although KRAS mutations are essential for tumorigenesis, as demonstrated in various mouse models, some human cancer cell lines harboring KRAS mutations exhibit no dependence on the KRAS protein for growth and survival, as shown by suppressing KRAS expression through CRISPR or siRNA techniques. Our group has previously reported that in a mouse model with inducible KRAS expression, specifically in the pancreas (via doxycycline administration), some tumors recur even with prolonged KRAS inhibition [65]. In contrast, cell lines that exhibit non-dependence on their driver mutations prior to inhibitor exposure are rare in EGFR-mutant or ALK-positive lung cancers and appear to be a unique feature of KRAS-mutant cancers. These KRAS-mutant cells may acquire non-dependence on mutant KRAS by activating alternative molecular pathways during tumor evolution, exhibiting primary resistance to KRAS inhibitors. Known mechanisms for acquiring non-dependence include YAP signaling activation, epithelial–mesenchymal transition (EMT), and KEAP1/NRF2 mutations [65,66,67,68]. YAP1 has been identified as a key driver of KRAS independence through regular screening (introducing open reading frame libraries followed by examining cDNA expression in surviving cells after KRAS inhibition) [69]. Our group has also shown that YAP1 bypasses KRAS function upon KRAS inhibition. Additionally, in mesenchymal marker-positive KRAS^G12C^ cell lines, PI3K signaling activation confers resistance to KRAS^G12C^ inhibitors. KEAP1/NRF2 signaling, which is responsible for maintaining cellular homeostasis via the induction of stress-responsive genes, also plays a role; approximately 20% of lung cancer cases harbor KEAP1 mutations, which correlate with lower response rates to sotorasib or adagrasib [70]. These mechanisms of KRAS independence are likely to interact in a complex manner.

3.Organ-Specific Sensitivity to KRAS Inhibitors

The efficacy of KRAS^G12C^ inhibitors varies depending on the tumor origin. For example, the response rate in lung cancer is approximately 40%, whereas in colorectal cancer, it is around 10–20%. In pancreatic cancer, though data is limited, the response rate is approximately 20%. Variability in response rates across different organs, with particularly low efficacy in colorectal cancer, has also been observed. In KRAS^G12C^-mutant colorectal cancer, EGFR feedback activation occurs after KRAS inhibition. Consequently, the efficacy of combination therapy with KRAS^G12C^ and EGFR inhibitors has been demonstrated in mouse models, with early clinical trials showing a response rate of 43% (12/28 patients) with combination therapy, compared to 22% (10/45 patients) with KRAS^G12C^ inhibitor monotherapy [71].

Similar feedback activation has been observed in BRAF-mutant colorectal cancer, where inhibition of mutant BRAF induces EGFR feedback activation. Thus, current therapies for BRAF-mutant colorectal cancer involve a combination of anti-EGFR antibodies, BRAF inhibitors, and MEK inhibitors. The importance of EGFR in feedback mechanisms is evident in RAS/RAF-mutant colorectal cancers. In KRAS-mutant lung cancer, EGFR, FGFR, and other receptor activations are also observed following KRAS^G12C^ inhibitor administration [64], but these do not necessarily contribute to primary resistance as seen in colorectal cancer. Organ-specific dependency on mutant KRAS and differences in signaling networks likely explain these variations, warranting further investigation.

#### 5.3.2. Acquired Resistance

Acquired resistance refers to cases where gene-targeted therapy initially shows efficacy but gradually loses effectiveness over time, ultimately leading to resistance. Mechanisms of acquired resistance in molecular-targeted therapies include epithelial–mesenchymal transition (EMT) and secondary mutations in the target gene.

It has been reported that KRAS^G12C^ inhibitors induce EMT, leading to adaptive resistance through the nuclear translocation of YAP, which induces MRAS, a member of the RAS superfamily, downstream of YAP [72]. In addition to the findings [72], recent studies have underscored the pivotal role of the YAP/TEAD pathway in resistance to KRAS G12C inhibitors. Functional genome-wide CRISPR screening revealed that KRAS G12C inhibitors activate RHOA signaling, which drives YAP nuclear localization and enhances tumor survival via YAP/TEAD activity. Notably, inhibiting TEAD has been shown to synergistically enhance the efficacy of KRAS G12C inhibitors [73].

Additionally, CRISPR-based studies have demonstrated the involvement of the PI3K pathway in acquired resistance. PI3K pathway alterations, such as PTEN loss, enable tumors to bypass KRAS G12C/SHP2 inhibition and sustain growth. PI3K inhibitors have shown promise in preclinical models to overcome these resistance mechanisms [34].

In terms of secondary mutations in the target gene, mutations in H95 and Y96, which are the key binding sites on mutant KRAS for the inhibitors, are most common [74]. Sotorasib and adagrasib both form covalent bonds with KRAS^G12C^, enhancing their binding affinity via non-covalent interactions with R68, H95, and Y96. Structural analyses suggest that adagrasib interacts with all three residues, while sotorasib primarily interacts with R68 and Y96, forming weaker bonds with H95 [74]. Therefore, adagrasib is susceptible to resistance from mutations in any of these residues, while sotorasib’s efficacy remains unaffected by H95 mutations. Additional bypass signaling mechanisms have been reported, including gene amplification of receptor tyrosine kinases (RTKs) such as EGFR, FGFR, and MET, and mutations in downstream components of the MAPK pathway. Our group also found that prolonged KRAS^G12C^ inhibition leads to the activation of multiple RTKs via YAP1, particularly activating the PI3K-AKT-mTOR pathway, thereby promoting tumor survival and resistance (*in press*). The diversity of these resistance mechanisms suggests a link with the unique biological characteristics of KRAS-mutant tumors as discussed in the section on primary resistance (Figure 4).

Drug resistance is more likely to be induced when the inhibitor’s activity is insufficient. However, our ability to increase the dose of currently available drugs is limited due to toxicity concerns. Therefore, the development of inhibitors with a higher selectivity for mutant KRAS may help circumvent resistance.

## 6. Future Strategies and Perspectives

### 6.1. Overcoming Resistance with Combination Therapy Using KRAS^G12C^ Inhibitors

As discussed above, resistance to KRAS inhibitors is a significant challenge in colorectal cancer. To address this, research is ongoing into combination therapies that pair KRAS inhibitors with agents targeting the mechanisms responsible for resistance development. KRAS^G12C^ inhibitors, which tend to have relatively mild side effects (such as diarrhea and nausea) and limited efficacy as monotherapies, are currently being tested in numerous clinical trials in combination with other agents, such as CDK4/6 inhibitors and mTOR inhibitors. Regulation of feedback mechanisms is crucial for enhancing the efficacy of KRAS^G12C^ inhibitors, especially in terms of overcoming both primary and acquired resistance.

In lung cancer, multiple receptors are activated, prompting clinical trials exploring combination therapies with SOS1 or SHP2 inhibitors to block the signaling pathway from receptors to KRAS. Additionally, mutated KRAS induces immunosuppressive factors (such as TGF-β and IL-6) via downstream effector signaling, negatively modulating the tumor immune microenvironment. While the effects of sotorasib on human cancer cell line-derived tumors in immunodeficient mice were only temporary, significant tumor regression was observed in syngeneic mouse models with intact immune systems. This suggests that sotorasib not only exerts a direct effect on tumor cells but may also enhance antitumor activity through immune cell induction. Based on these findings, clinical trials are also underway investigating the combination of sotorasib with anti-PD-1 antibodies.

### 6.2. Development of Targeted Therapies for Mutations Beyond KRAS^G12C^

#### 6.2.1. KRAS^G12D^ Inhibitor

While clinical applications of KRAS inhibitors have initially focused on G12C, efforts are underway to develop inhibitors for other mutation sites. G12D inhibitors are currently in clinical trials, both as a monotherapy and in combination with EGFR inhibitors, and inhibitors for mutations such as G12V have shown promising in vitro results.

In non-small cell lung cancer (NSCLC), the G12C mutation is the most prevalent KRAS mutation, accounting for 41% of all mutations. However, this mutation is relatively rare in gastrointestinal cancers, occurring in only 7% of colorectal cancers and 1% of pancreatic cancers, thereby limiting the patient population that may benefit from G12C inhibitors in these cancer types. In contrast, the G12D mutation is the most common KRAS mutation in gastrointestinal cancers, found in 28% of colorectal cancers and 39% of pancreatic cancers (Table 2). This underscores the significant clinical relevance of developing G12D inhibitors [75,76].

A major breakthrough in developing KRAS-specific inhibitors was the discovery of the switch II pocket within the KRAS protein. However, for practical applications, it was not sufficient for the compounds to merely bind to this pocket. The development of G12C inhibitors requires compounds that can stably form a covalent bond with the cysteine residue, allowing for the irreversible inhibition of KRAS-GDP. In contrast, with the G12D mutation, which involves an aspartate substitution, no method has been established to create compounds that form covalent bonds with aspartate. As a result, KRAS G12D inhibitors must rely on non-covalent (reversible) inhibition.

Furthermore, the KRAS G12C mutation retains greater inactivation with KRAS-GDP, allowing for the successful development of GDP-trapping agents. However, the intrinsic GTPase activity of KRAS G12D is only about 40% of that of KRAS G12C. Even if KRAS-GDP trapping was feasible, the presence of active KRAS-G12D (KRAS-GTP) would likely limit the drug’s efficacy. Therefore, it is uncertain whether a GDP-trapping inhibitor for KRAS G12D would achieve significant efficacy in tumor cells.

To address these challenges, Mirati Therapeutics conducted high-resolution crystal structure analyses of compounds bound to mutant KRAS proteins, optimizing these compounds to develop MRTX1133, a non-covalent, selective inhibitor for KRAS G12D [60]. While it remains unknown whether G12D inhibitors will produce effects comparable to G12C inhibitors, foundational studies have confirmed their efficacy, and clinical trials are underway. Studies on resistance mechanisms have also begun, revealing that, similar to G12C inhibitors, feedback signaling activates EGFR, with reports indicating the combinatorial benefit of MRTX1133 with anti-EGFR antibodies.

Unlike sotorasib, which acts as a KRAS OFF inhibitor, MRTX1133 is a KRAS ON inhibitor. Nonetheless, MRTX1133 has demonstrated selective inhibitory effects on the growth of colorectal and pancreatic cancer cells, highlighting its potential for future clinical application (Figure 1). Currently, G12D inhibitors differ from G12C inhibitors in that their low oral bioavailability necessitates intravenous administration [75,77].

#### 6.2.2. G12X Inhibitors and SOS1 Inhibitors

Even if combination therapies centered on KRAS G12D and G12C inhibitors are established, they remain ineffective against the more frequent G12V and G13D mutations in colorectal and pancreatic cancers. This has led to interest in novel therapies, such as the G12X inhibitor (RMC-6236), which targets all G12 codon mutations, and the SOS1 inhibitor (BI-3405), which is designed to target all KRAS mutations (Figure 2). The SOS1–KRAS interaction inhibitor BI-3406 has shown sensitivity in lung, colorectal, and pancreatic cancer cells with KRAS G12V, G12A, and G13D mutations, with enhanced effects when combined with MEK inhibitors [78].

Moreover, RMC-6236, a KRAS ON inhibitor that binds to an active pocket that is common to all KRAS 12 codon mutations and inhibits its activity, was highlighted at the 2022 American Association for Cancer Research (AACR) meeting. While the efficacy of G12X and SOS1 inhibitors shows promise, resistance signaling is expected to emerge, and it is likely that these inhibitors may not be universally effective across all cancer types.

### 6.3. KRAS Inhibitory Peptides

Sakamoto et al. have developed KS-58, a KRAS G12D-selective inhibitory peptide, which has gained considerable attention [79,80]. Additionally, Chugai Pharmaceutical has independently constructed a peptide library containing non-natural amino acids aimed at mid-molecule drug discovery. Through screening with this library, they identified an orally available cyclic peptide, LUNA-18, which inhibits the protein–protein interaction between RAS and RAS GEF, keeping RAS in an inactive state. LUNA-18 is capable of binding to various forms of RAS, including the wild-type, and is currently in clinical trials (NCT05012618).

### 6.4. Anti-RAS Vaccines

Another novel therapeutic approach involves vaccines targeting KRAS proteins to induce an immune response. The GI-4000 series is composed of four heat-inactivated recombinant *S. cerevisiae* yeast strains, which have been shown in preclinical models to induce tumor reductions [81,82]. Additionally, GI-4000 demonstrated a favorable safety and immunologic profile in most colorectal cancer (CRC) patients in a phase I clinical trial [67].

Another strategy, combining RAS peptides with granulocyte-macrophage colony-stimulating factor (GM-CSF), developed by Targovax, has been shown to elicit T-cell-mediated antitumor immune responses against mutant RAS peptides [83]. Targovax’s second-generation vaccine, TG02, is currently under investigation in a phase Ib clinical trial for CRC, both as a monotherapy and in combination with immune checkpoint inhibitors.

In addition, mRNA vaccines coding for novel epitopes of common KRAS mutations are being developed to stimulate T-cell responses against these mutant KRAS neoepitopes. The safety and tolerability of one such mRNA vaccine, mRNA-5671, are currently being evaluated in a phase I clinical trial involving patients with KRAS-mutant tumors (NSCLC, CRC, and pancreatic adenocarcinoma), either as a monotherapy or in combination with pembrolizumab (NCT03948763).

### 6.5. Targeting KRAS-Driven Metabolic Adaptations

Oncogenic KRAS extensively reprograms cancer cell metabolism to sustain proliferation and survival. This includes enhanced glycolysis, glutaminolysis, and aspartate metabolism, driven by the upregulation of key enzymes such as PCK1, PCK2, and ASS1. These metabolic adaptations provide critical precursors for anabolic processes and redox balance. Targeting these pathways has shown promise in preclinical studies. For example, glutaminase inhibitors and agents targeting aspartate metabolism, such as ASS1 inhibitors, may enhance the efficacy of KRAS inhibitors in colorectal cancer [33,35].

### 6.6. Development of Therapies Using Oligonucleotide Therapeutics

Oligonucleotide therapeutics, often referred to as nucleic acid medicines, are typically synthesized chemically and may include various modifications to their sugar, base, or backbone structures to improve their stability and pharmacological properties. Notable examples include antisense oligonucleotides (ASOs), RNA interference (RNAi) agents, and aptamers. Unlike gene therapies, which introduce specific DNA genes into the body to induce mRNA expression and subsequent protein production, nucleic acid therapeutics act directly on targets like mRNA [84].

Recent discoveries have revealed new oncogenic mechanisms and vulnerabilities associated with RAS gene mutations, leading to reports of innovative treatments using nucleic acid therapeutics. Kobayashi et al. identified that silent mutations (genetic mutations that do not alter the amino acid sequence of the protein) are essential for the oncogenicity of the KRAS Q61K mutation. This discovery highlighted that the regions surrounding codon Q61 in KRAS, NRAS, and HRAS are particularly susceptible to splicing errors. Building on these findings, researchers have explored the potential of inducing the body’s natural splicing mechanisms with nucleic acid therapeutics, offering a novel therapeutic approach that selectively targets cancer cells harboring oncogenic mutations [84].

## 7. Conclusions

KRAS mutations lie at the core of colorectal cancer (CRC) pathogenesis, driving persistent activation of the MAPK/ERK and PI3K/AKT pathways and conferring resistance to numerous therapies, particularly anti-EGFR antibodies. While recently developed KRAS inhibitors, such as sotorasib and adagrasib, mark an important milestone—especially for KRASG12C-mutated cancers—their efficacy in CRC remains limited. Novel strategies must therefore address both primary and acquired resistance mechanisms.

Combination Therapies Targeting Feedback Pathways Preclinical and early clinical data suggest that the co-targeting of EGFR or other receptor tyrosine kinases can overcome adaptive resistance to KRAS inhibition, particularly in CRC. In parallel, the concomitant blockade of the PI3K/AKT or MAPK/MEK pathways may further enhance tumor response and delay resistance.Expanding the Repertoire of KRAS Mutant–Specific Agents Emerging efforts are focused on KRAS^G12D^, ^G12V^, and ^G13D^ inhibitors to meet the needs of patients harboring these mutations. These next-generation compounds, including non-covalent inhibitors and cyclic peptides, will require rigorous validation in CRC-specific models to ensure clinical efficacy.Liquid Biopsy–Guided Precision Medicine The routine implementation of minimally invasive techniques for circulating tumor DNA (ctDNA) analyses will enable repeated, real-time monitoring of RAS mutation status. This approach can detect emerging resistance mutations early, guide treatment decisions, and facilitate therapy adaptations before clinical progression occurs, representing a crucial step toward effective, personalized KRAS-targeted interventions.Exploiting Novel Vulnerabilities Promising avenues include macropinocytosis inhibitors, antisense oligonucleotides, and tumor vaccines to disrupt RAS-driven nutrient uptake and immunosuppression. Such strategies may synergize with or replace direct KRAS inhibition, especially in tumors that exhibit alternative resistance pathways or limited KRAS dependence.

Moving forward, a personalized treatment paradigm will hinge on deeper molecular profiling, strategic combination regimens, and continuous surveillance through liquid biopsies. Although KRAS mutations have historically been labeled as “undruggable”, the rapid evolution of therapeutic platforms—encompassing small molecules, RNA-based drugs, and immunotherapies—holds the potential to transform outcomes for patients with KRAS-mutant CRC. Achieving this goal will demand sustained interdisciplinary efforts linking fundamental discoveries to well-designed clinical trials, ultimately translating into more durable and effective therapies.

## Figures and Tables

**Figure 1 cancers-17-00428-f001:**
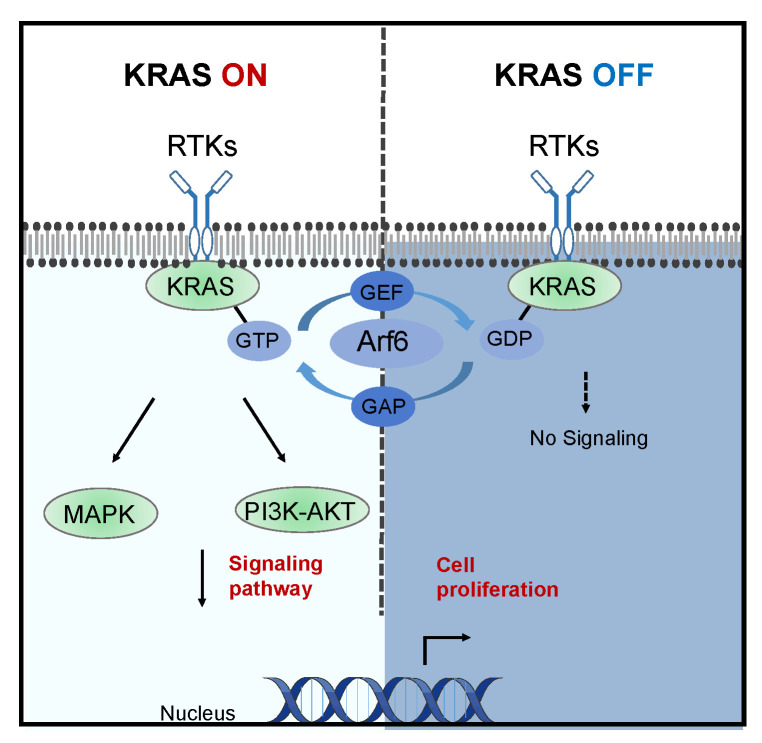
Activation states of KRAS and their impact on signaling pathways. This schematic illustrates the activation cycle of KRAS, highlighting its transition between the inactive GDP-bound state (“KRAS OFF”) and the active GTP-bound state (“KRAS ON”). In the GDP-bound state, KRAS remains inactive, with no downstream signaling. Upon receptor tyrosine kinase (RTK) activation, guanine nucleotide exchange factors (GEFs) facilitate the exchange of GDP for GTP, activating KRAS. Active KRAS (GTP-bound) initiates multiple downstream signaling pathways, including the MAPK and PI3K-AKT cascades, driving cell proliferation. GTPase-activating proteins (GAPs) subsequently inactivate KRAS by hydrolyzing GTP back to GDP. Additionally, Arf6-mediated mechanisms in KRAS regulation are depicted. This diagram emphasizes the functional significance of KRAS’s activation states in regulating cellular signaling and proliferation.

**Figure 2 cancers-17-00428-f002:**
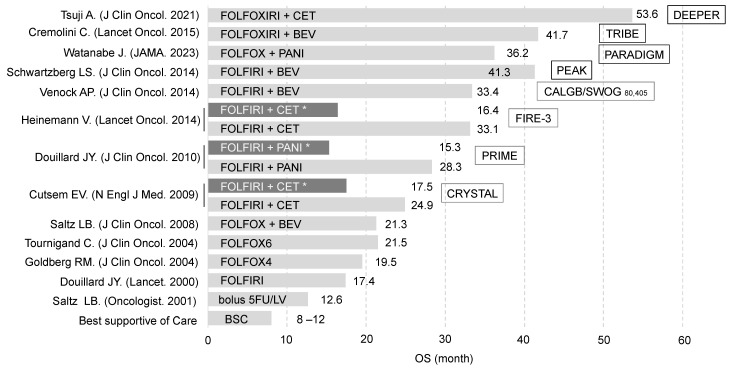
Evolution of chemotherapy and survival rates in unresectable advanced and recurrent colorectal cancer. Light gray represents cases with wild-type KRAS, while dark gray (*) indicates cases with KRAS mutations. Due to differences in time periods and study backgrounds, direct comparisons of survival rates are not feasible.

**Figure 3 cancers-17-00428-f003:**
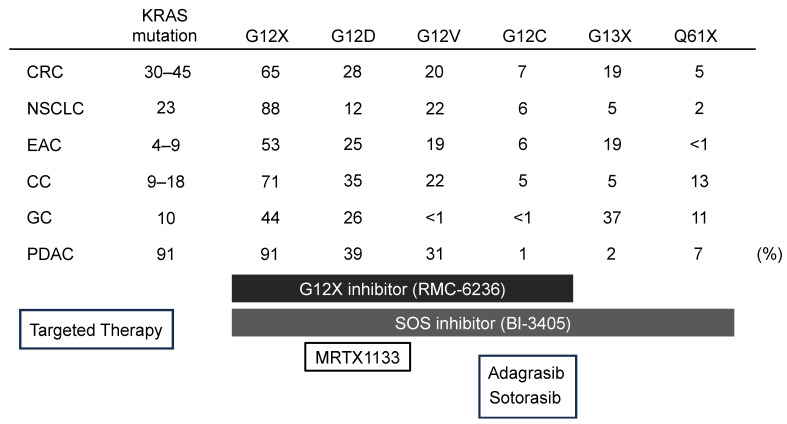
KRAS mutation rates across various cancer types and corresponding targeted therapies. The upper figure depicts the proportion (%) of KRAS codons across various cancer types, while the lower figure highlights the types of targeted therapies currently available for each codon. CRC (colorectal cancer), NSCLC (non-small cell lung cancer), EAC (esophageal adenocarcinoma), CC (cholangiocarcinoma), GC (gastric cancer), PDAC (pancreatic ductal adenocarcinoma).

**Figure 4 cancers-17-00428-f004:**
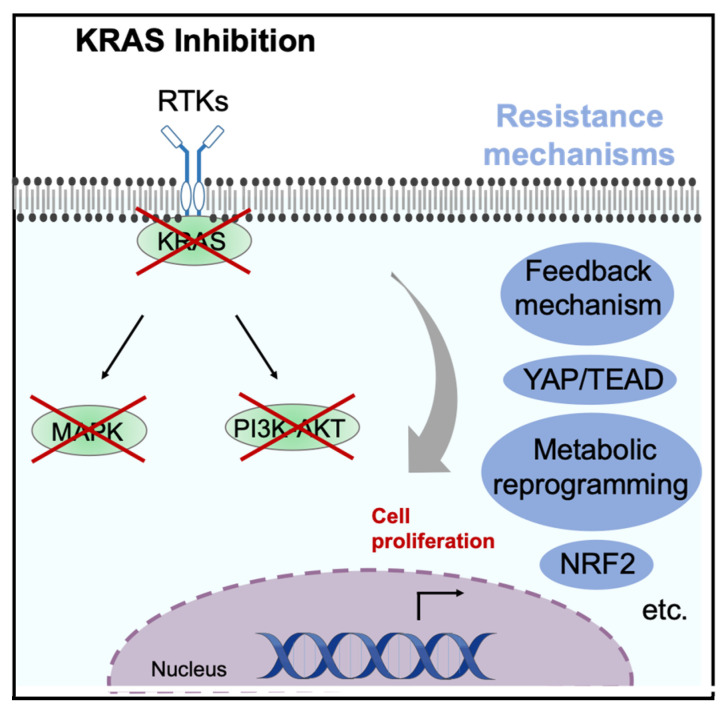
The resistance mechanism in colorectal cancer during KRAS inhibition. This model depicts the key resistance mechanisms. This figure also highlights pathway crosstalk, including the role of metabolic reprogramming and the activation of compensatory signaling nodes (e.g., YAP/TEAD and NRF2).

**Table 1 cancers-17-00428-t001:** Proportion of KRAS exon 2 mutations by each stage.

	Stage	Proportion		Dukes’ Stage	Proportion
Watanabe, et al. [45] N = 5887	I	33.1%	Andreyev, et al. [46] N = 2721	A	33.9%
	II	37.3%		B	39.8%
	III	38.1%		C	38.3%
	IV	37.5%		D	35.8%

**Table 2 cancers-17-00428-t002:** Proportion of genetic mutations for each KRAS exon and measurement methods.

Clinical Trial	Exon 2	Exon 3	Exon 4	Exon 22	Exon 33	Exon 44	Methods
PRIME (n = 1096, %)	40.1	4.5	5.8	3.5	4.4	0.0	SURVEYOR
20050181 (n = 1083, %)	44.9	4.4	7.7	2.2	5.6	0.0	SURVEYOR
20020408 (n = 427, %)	43.0	4.8	5.0	4.2	3.0	1.1	SURVEYOR
OPUS (n = 315, %)	43.0	4.9	9.3	6.8	5.1	0.8	BEAMing
CRYSTAL (n = 1198, %)	37.0	3.3	5.6	3.5	2.8	0.9	BEAMing
PEAK (n = 225, %)	N/A	4.0	7.0	5.0	6.0	0.0	SURVEYOR
FIRE-3 (n = 468, %)	N/A	4.0	5.9	3.6	2.1	0.2	Pyrosequence
CALGB80405 (n = 1137, %)	1.3	4.0	5.9	2.3	4.2	0.0	BEAMing
RASKET (n = 307, %)	37.8	2.0	3.3	2.0	2.6	0.0	Luminex

## Data Availability

The data supporting the findings of this study can be obtained from the corresponding authors, upon reasonable request.

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
