# Peer review of "The Role of KRAS Mutations in Colorectal Cancer: Biological Insights, Clinical Implications, and Future Therapeutic Perspectives"

_cancers, 2025, doi:10.3390/cancers17030428_

Round 1
Reviewer 1 Report
Comments and Suggestions for Authors
The review attempts to dissect the intricacies of the KRAS mutation. Reasonably well written but it has focused only on two mutations for which small molecule inhibitors are available (G12C and G12D). What about the other 5 (G12V, G12S, G12R, G12A and G13D) commonly found mutations of codon 12 and 13. Some discussions are solicited.
Please explain in some greater detail and possibly with a cartoon or illustrations the on inhibitor, off inhibitor concepts (lines 496-497) for better understanding of the readers not working with KRAS.
Please discuss the overall structural features of KRAS protein and all the loops and switches rather than switch II pocket only. Please put in a couple of sentences as to why the authors think that KRAS was undruggable for over a decade?
Author Response
Comments ①:The review attempts to dissect the intricacies of the KRAS mutation. Reasonably well written but it has focused only on two mutations for which small molecule inhibitors are available (G12C and G12D). What about the other 5 (G12V, G12S, G12R, G12A and G13D) commonly found mutations of codon 12 and 13. Some discussions are solicited.
Response ①: We appreciate the reviewer’s point regarding the need to address other recurrent mutations in KRAS beyond G12C and G12D. The following text has been added to line 116.
The most frequently detected mutations in codons 12 and 13—namely G12V (the incidence; 20-25%), G12S (5-7%), G12R (3〜5%), G12A (3〜5%), and G13D (15-20%)—are also clinically relevant, with prevalence varying by tumor type. While G12C and G12D have garnered significant attention due to the development of novel small molecule inhibitors, other substitutions remain relatively underexplored in terms of direct targeted therapies.
For example, G12V is frequently identified in pancreatic ductal adenocarcinomas and is associated with aggressive disease biology. G12R, which is also common in pancreatic cancer, displays altered biochemical properties—particularly in GDP/GTP hydrolysis—suggesting that unique conformational features may influence inhibitor design. Moreover, G12S and G12A occur in both colorectal and lung cancers at lower but still significant frequencies. These variants often retain high GTP affinity and can activate downstream effectors in a manner subtly distinct from G12C or G12D. Finally, G13D—while less frequently discussed—represents a non-negligible subset of RAS mutations, especially in colorectal cancer, where it may respond differently to anti-EGFR therapies compared with other mutations.
Comments ②: Please explain in some greater detail and possibly with a cartoon or illustrations the on inhibitor, off inhibitor concepts (lines 496-497) for better understanding of the readers not working with KRAS.
Response ②: We appreciate the reviewer's suggestion to elaborate on the concepts of "On inhibitors" and "Off inhibitors" to enhance the manuscript's accessibility to readers who may not be familiar with KRAS biology. To improve understanding, we have included a schematic diagram (Figure 1) illustrating the KRAS activation cycle and the points of intervention for "On" and "Off" inhibitors. We believe this addition will make the manuscript more accessible to a broader audience and thank the reviewer for this constructive suggestion.
Comments ③:Please discuss the overall structural features of KRAS protein and all the loops and switches rather than switch II pocket only.
Response ③:We appreciate your suggestion to broaden the discussion on KRAS beyond the Switch II pocket. The following text has been added to line 92.
Structurally, KRAS belongs to the small GTPase family and contains a highly conserved N-terminal G-domain of approximately 170 residues and a C-terminal Hypervariable region (HVR) of approximately 25 residues. This G-domain is composed of five alpha-helices and six beta-strands, and forms a catalytic core responsible for GTP binding and hydrolysis. There are several important regions in this domain.
- P-loop (phosphate-binding loop): This loop (residues 10-16) stabilizes the phosphate of GTP or GDP.
- Switch I (residues 32-38): This region undergoes a conformational change during GTP hydrolysis and is essential for interaction with effectors such as RAF.
- Switch II (residues 59-67): This region adopts a different conformation in the GDP-bound and GTP-bound states. Although this region has been the focus of much drug discovery research in recent years, it is not the only region important for KRAS function.
- Additional loops and regions: Beyond switch I and II, other structural elements regulate KRAS's affinity for nucleotides and its ability to bind various regulators and effectors (e.g., guanine nucleotide exchange factors, GEFs, and GTPase activating proteins, GAPs).  
Genetic analysis has shown that 98% of RAS mutations that cause cancer are concentrated in three residues: G12, G13, and Q61. All of these are located near the phosphate group of GTP (p-loop: G12 & G13, Switch II: Q61), and it is thought that mutations in these residues stabilize the GTP-bound form and cause constant signal activation.
Comments ④:Please put in a couple of sentences as to why the authors think that KRAS was undruggable for over a decade?
Response ④:We appreciate your suggestion to broaden the discussion that KRAS was undruggable for over a decade. The following text has been added to line 234.
“Due to issues such as the binding affinity between GTP and RAS proteins and the structure of the proteins, KRAS was undruggable for over a decade.” “These features collectively contributed to the long-held view that directly targeting KRAS was beyond reach.”

Reviewer 2 Report
Comments and Suggestions for Authors
The manuscript provides a comprehensive overview of KRAS mutations as a key factor in colorectal cancer, highlighting their impact on prognosis and treatment choices. Overall, the manuscript offers valuable insights into the biological and clinical implications of KRAS mutations.
In the Introduction, the authors initially refer to colon cancer before shifting to colorectal cancer.
Given that several KRAS mutations beyond codons 12 and 13 are clinically relevant, albeit less frequent, Section 2 would benefit from a schematic overview of these mutations, including their incidence and impact on protein function. Although some of these mutations are mentioned in Section 5, a more structured presentation in Section 2 would be helpful.
The recent findings regarding the functions of KRAS, discussed in Section 3, are highly relevant and should be explored in greater detail, either in this section or elsewhere in the manuscript, with respect to current and emerging therapeutic strategies. It should also be clarified which functions are affected by KRAS mutations, as it seems that most of the roles discussed pertain to the mutant protein. If this is the case, the section titles should specifically reference "KRAS mutations" rather than just "KRAS."
The authors should remove references to publications "in press," and, considering this, Section 6.5 should be excluded.
In Section 6.6, the term "chemically modified nucleotides" may need reconsideration, as oligonucleotide therapeutics do not contain modified nucleotides by definition.
The conclusion remains overly broad and generalized. It would be more valuable for the authors to provide specific recommendations regarding which molecular processes and therapeutic approaches show the most promise for future research, particularly from a clinical perspective.
Author Response
Comments ①:The manuscript provides a comprehensive overview of KRAS mutations as a key factor in colorectal cancer, highlighting their impact on prognosis and treatment choices. Overall, the manuscript offers valuable insights into the biological and clinical implications of KRAS mutations. In the Introduction, the authors initially refer to colon cancer before shifting to colorectal cancer.
Response ①:We appreciate your suggestion. We have revised the term "colon cancer" to "colorectal cancer" in the introduction as suggested.
Comments ②:Given that several KRAS mutations beyond codons 12 and 13 are clinically relevant, albeit less frequent, Section 2 would benefit from a schematic overview of these mutations, including their incidence and impact on protein function. Although some of these mutations are mentioned in Section 5, a more structured presentation in Section 2 would be helpful.
Response ②:We sincerely thank the reviewer for their valuable suggestion to provide a schematic overview of KRAS mutations beyond codons 12 and 13 in Section 2. We agree that a more structured presentation, including the incidence and functional impact of these mutations, would enhance the manuscript's clarity and provide a more comprehensive understanding for readers.
The following text has been added to line 116.
The most frequently detected mutations in codons 12 and 13—namely G12V (the incidence; 20-25%), G12S (5-7%), G12R (3〜5%), G12A (3〜5%), and G13D (15-20%)—are also clinically relevant, with prevalence varying by tumor type. While G12C and G12D have garnered significant attention due to the development of novel small molecule inhibitors, other substitutions remain relatively underexplored in terms of direct targeted therapies.
For example, G12V is frequently identified in pancreatic ductal adenocarcinomas and is associated with aggressive disease biology. G12R, which is also common in pancreatic cancer, displays altered biochemical properties—particularly in GDP/GTP hydrolysis—suggesting that unique conformational features may influence inhibitor design. Moreover, G12S and G12A occur in both colorectal and lung cancers at lower but still significant frequencies. These variants often retain high GTP affinity and can activate downstream effectors in a manner subtly distinct from G12C or G12D. Finally, G13D—while less frequently discussed—represents a non-negligible subset of RAS mutations, especially in colorectal cancer, where it may respond differently to anti-EGFR therapies compared with other mutations.
Comments â‘¢The recent findings regarding the functions of KRAS, discussed in Section 3, are highly relevant and should be explored in greater detail, either in this section or elsewhere in the manuscript, with respect to current and emerging therapeutic strategies.
Response ③:We are grateful to the reviewer for highlighting the relevance of the recent findings regarding KRAS functions discussed in Section 3 and for encouraging a deeper exploration of their implications for current and emerging therapeutic strategies. We fully agree that integrating these insights enhances the manuscript's depth and clinical relevance.
To address this, we have expanded the discussion on the metabolic reprogramming driven by oncogenic KRAS, particularly in the context of colorectal cancer. Specifically, we have incorporated the following text into Section 3 (3.5 Rewiring of Cancer Metabolism by Oncogenic KRAS):
"Oncogenic KRAS extensively reprograms cancer cell metabolism to sustain proliferation and survival. This includes enhanced glycolysis, glutaminolysis, and aspartate metabolism, driven by upregulation of key enzymes such as PCK1, PCK2, and ASS1. These metabolic adaptations provide critical precursors for anabolic processes and redox balance. Targeting these pathways has shown promise in preclinical studies. For example, glutaminase inhibitors and agents targeting aspartate metabolism, such as ASS1 inhibitors, may enhance the efficacy of KRAS inhibitors in colorectal cancer (Mukhopadhyay et al., 2021; Doubleday et al., 2022)."
We believe this addition underscores the therapeutic potential of targeting metabolic vulnerabilities in KRAS-mutant colorectal cancer and aligns well with the manuscript's focus. Your insightful comments have significantly enriched the manuscript, and we are confident that these revisions will enhance its impact and relevance. Thank you once again for your thoughtful suggestions.
Comments ④:It should also be clarified which functions are affected by KRAS mutations, as it seems that most of the roles discussed pertain to the mutant protein. If this is the case, the section titles should specifically reference "KRAS mutations" rather than just "KRAS."
Response ④:We appreciate your suggestion. We have revised the term "KRAS" to " KRAS mutations " in the section titles as suggested.
Comments ⑤:The authors should remove references to publications "in press," and, considering this, Section 6.5 should be excluded.
Response ⑤:We appreciate your comment regarding the references to publications "in press" and their suggestion to exclude Section 6.5. Section 6.5 has been excluded in its entirety. And We changed to a completely different item. We thank the reviewer for highlighting this issue.
Comments ⑥:In Section 6.6, the term "chemically modified nucleotides" may need reconsideration, as oligonucleotide therapeutics do not contain modified nucleotides by definition.
Response ⑥:Thank you for your valuable feedback regarding our use of the term “chemically modified nucleotides” in Section 6.6. We understand that some definitions restrict “oligonucleotide therapeutics” to unmodified sequences, and we appreciate your suggestion to reconsider our terminology. To address this concern, we propose the following revisions:
- Clarification in the Manuscript We will adjust our language to distinguish between “oligonucleotide therapeutics” as a broad category and “chemically modified oligonucleotide therapeutics” as a commonly used subclass. In doing so, we can highlight that while many clinically utilized oligonucleotide therapeutics do employ chemical modifications (e.g., phosphorothioate backbone, 2’-O-modifications, locked nucleic acids) to enhance stability and efficacy, the term “oligonucleotide therapeutics” in its purest definition can refer to unmodified sequences.
- Terminology Update In Section 6.6, we will rephrase our sentence to read: “Oligonucleotide therapeutics, often referred to as nucleic acid medicines, are typically synthesized chemically and may include various modifications to their sugar, base, or backbone structures to improve stability and pharmacological properties. ”This revision acknowledges both the classical definition (no modifications) and the current practical reality (frequent use of chemical modifications).
Comments ⑦:The conclusion remains overly broad and generalized. It would be more valuable for the authors to provide specific recommendations regarding which molecular processes and therapeutic approaches show the most promise for future research, particularly from a clinical perspective
Response ⑦:Thank you for your insightful comment regarding the breadth of our conclusion. We agree that providing more specific recommendations on molecular processes and therapeutic strategies would enhance the clinical relevance of this section. We propose the following revisions to address your concerns:
KRAS mutations lie at the core of colorectal cancer (CRC) pathogenesis, driving persistent activation of the MAPK/ERK and PI3K/AKT pathways and conferring resistance to numerous therapies, particularly anti-EGFR antibodies. While recently developed KRAS inhibitors, such as sotorasib and adagrasib, mark an important milestone—especially for KRASG12C-mutated cancers—their efficacy in CRC remains limited. Novel strategies must therefore address both primary and acquired resistance mechanisms. Specifically:
- Combination Therapies Targeting Feedback Pathways Preclinical and early clinical data suggest that co-targeting of EGFR or other receptor tyrosine kinases can overcome adaptive resistance to KRAS inhibition, particularly in CRC. In parallel, concomitant blockade of the PI3K/AKT or MAPK/MEK pathways may further enhance tumor response and delay resistance.
- Expanding the Repertoire of KRAS Mutant–Specific Agents Emerging efforts are focused on KRAS^G12D, G12V, and G13D inhibitors to meet the needs of patients harboring these mutations. These next-generation compounds, including non-covalent inhibitors and cyclic peptides, will require rigorous validation in CRC-specific models to ensure clinical efficacy.
- Liquid Biopsy–Guided Precision Medicine The routine implementation of minimally invasive techniques for circulating tumor DNA (ctDNA) analyses will enable repeated, real-time monitoring of RAS mutation status. This approach can detect emerging resistance mutations early, guide treatment decisions, and facilitate therapy adaptations before clinical progression, representing a crucial step toward effective, personalized KRAS-targeted interventions.
- Exploiting Novel Vulnerabilities Promising avenues include macropinocytosis inhibitors, antisense oligonucleotides, and tumor vaccines to disrupt RAS-driven nutrient uptake and immunosuppression. Such strategies may synergize with or replace direct KRAS inhibition, especially in tumors that exhibit alternative resistance pathways or limited KRAS dependence.
Moving forward, a personalized treatment paradigm will hinge on deeper molecular profiling, strategic combination regimens, and continuous surveillance through liquid biopsy. Although KRAS mutations have historically been labeled “undruggable,” the rapid evolution of therapeutic platforms—encompassing small molecules, RNA-based drugs, and immunotherapies—holds potential to transform outcomes for patients with KRAS-mutant CRC. Achieving this goal will demand sustained interdisciplinary efforts linking fundamental discoveries to well-designed clinical trials, ultimately translating into more durable and effective therapies.

Reviewer 3 Report
Comments and Suggestions for Authors
Dr. Takada and the group have authored a timely review article exploring the role of KRAS mutations in colorectal cancer, with a focus on biological insights, clinical implications, and future therapeutic perspectives. This well-written review holds significant translational value. However, a few points need to be addressed before it is ready for acceptance:
1. In addition to Adachi's paper (PMID: 37277529), other studies have demonstrated that the YAP/TEAD pathway is a major factor contributing to resistance against KRAS G12C and KRAS G12C + SHP2 inhibitor combinations. Whole-genome CRISPR screens have also validated other key pathways, including PI3K and NRF2, alongside YAP/TEAD. The authors should add a line discussing this and reference these important works (PMID: 37729426 and PMID: 37934115) to ensure the review remains comprehensive and up to date.
2. Cancer metabolism plays a definitive role in oncogenic RAS-mediated signaling pathways, as well established in the literature (PMID: 33870211 and PMID: 34227245). Targeting cancer metabolism is a critical strategy for overcoming resistance to KRAS inhibitors in various cancers, including colorectal cancer. The authors should address this aspect, referencing the relevant studies to provide a more comprehensive perspective.
3. Finally, the authors should include a model illustrating the known signaling pathways regulating KRAS-mediated signal transduction in colorectal cancer, highlighting mechanisms of resistance and potential therapeutic interventions arising from pathway crosstalk. This model will encapsulate the central theme of the work.
Author Response
Dr. Takada and the group have authored a timely review article exploring the role of KRAS mutations in colorectal cancer, with a focus on biological insights, clinical implications, and future therapeutic perspectives. This well-written review holds significant translational value. However, a few points need to be addressed before it is ready for acceptance:
Comments ①:In addition to Adachi's paper (PMID: 37277529), other studies have demonstrated that the YAP/TEAD pathway is a major factor contributing to resistance against KRAS G12C and KRAS G12C + SHP2 inhibitor combinations. Whole-genome CRISPR screens have also validated other key pathways, including PI3K and NRF2, alongside YAP/TEAD. The authors should add a line discussing this and reference these important works (PMID: 37729426 and PMID: 37934115) to ensure the review remains comprehensive and up to date.
Response ①:We fully agree with your suggestion to include a discussion of the YAP/TEAD pathway and other key pathways identified in recent studies. In the revised manuscript, we have incorporated the following updates:
To ensure coherence, we integrated these updates into the "5.3.2. Acquired Resistance" section of the manuscript. The revised text now reads as follows:
"In addition to the findings by Adachi et al. (PMID: 37277529), recent studies have underscored the pivotal role of the YAP/TEAD pathway in resistance to KRAS G12C inhibitors. Functional genome-wide CRISPR screening revealed that KRAS G12C inhibitors activate RHOA signaling, which drives YAP nuclear localization and enhances tumor survival via YAP/TEAD activity. Notably, inhibiting TEAD has been shown to synergistically enhance the efficacy of KRAS G12C inhibitors (PMID: 37934115).
Additionally, CRISPR-based studies have demonstrated the involvement of the PI3K pathway in acquired resistance. PI3K pathway alterations, such as PTEN loss, enable tumors to bypass KRAS G12C/SHP2 inhibition and sustain growth. PI3K inhibitors have shown promise in preclinical models to overcome these resistance mechanisms (PMID: 37729426)."
We believe these additions address your comments and significantly enhance the discussion of acquired resistance mechanisms in the context of KRAS-mutant tumors.
Comments ②:Cancer metabolism plays a definitive role in oncogenic RAS-mediated signaling pathways, as well established in the literature (PMID: 33870211 and PMID: 34227245). Targeting cancer metabolism is a critical strategy for overcoming resistance to KRAS inhibitors in various cancers, including colorectal cancer. The authors should address this aspect, referencing the relevant studies to provide a more comprehensive perspective.
Response ②:Thank you for your valuable feedback and for emphasizing the importance of addressing the role of cancer metabolism in oncogenic RAS-mediated signaling pathways. We appreciate the opportunity to enhance our manuscript by integrating recent findings on this topic. Below, we provide a detailed response to your comments:
We agree that addressing the metabolic adaptations mediated by oncogenic KRAS is crucial to understanding resistance mechanisms and therapeutic vulnerabilities. To address this, we have incorporated additional discussion into the manuscript, referencing the two provided studies. Specifically:
We have included the following text in the revised section on 3-5
Rewiring of Cancer Metabolism by Oncogenic KRAS
Oncogenic KRAS significantly reprograms cellular metabolism, enhancing glycolysis, glutaminolysis, and aspartate metabolism to fuel tumor growth and survival. Recent findings have highlighted that KRAS mutations drive the upregulation of key enzymes, such as PCK1, PCK2, and ASS1, which are involved in aspartate and urea cycle metabolism, contributing to the metabolic flexibility of KRAS-mutant colorectal cancer (CRC) cells ​. These adaptations provide critical precursors for nucleotide biosynthesis and support redox homeostasis, making them attractive targets for therapeutic intervention.
We have included the following text in the revised section on 6. (6-5)
"Oncogenic KRAS extensively reprograms cancer cell metabolism to sustain proliferation and survival. This includes enhanced glycolysis, glutaminolysis, and aspartate metabolism, driven by upregulation of key enzymes such as PCK1, PCK2, and ASS1. These metabolic adaptations provide critical precursors for anabolic processes and redox balance. Targeting these pathways has shown promise in preclinical studies. For example, glutaminase inhibitors and agents targeting aspartate metabolism, such as ASS1 inhibitors, may enhance the efficacy of KRAS inhibitors in colorectal cancer (Mukhopadhyay et al., 2021; Doubleday et al., 2022)."
We sincerely thank you for your thoughtful suggestions, which have enriched our manuscript and broadened its scope. We are confident that these revisions address your comments and enhance the overall impact of our work.
Comments ③: Finally, the authors should include a model illustrating the known signaling pathways regulating KRAS-mediated signal transduction in colorectal cancer, highlighting mechanisms of resistance and potential therapeutic interventions arising from pathway crosstalk. This model will encapsulate the central theme of the work.
Response ③:We have added the following model as Figure 1 and 4 to the revised manuscript, along with a corresponding legend:
Figure1: Activation States of KRAS and Their Impact on Signaling Pathways
Figure 4: The resistance mechanism in colorectal cancer during KRAS inhibition
Figure1. Activation States of KRAS and Their Impact on Signaling Pathways. The schematic illustrates the activation cycle of KRAS, highlighting its transition between the inactive GDP-bound state ("KRAS OFF") and the active GTP-bound state ("KRAS ON"). In the GDP-bound state, KRAS remains inactive, with no downstream signaling. Upon receptor tyrosine kinase (RTK) activation, guanine nucleotide exchange factors (GEFs) facilitate the exchange of GDP for GTP, activating KRAS. Active KRAS (GTP-bound) initiates multiple downstream signaling pathways, including the MAPK and PI3K-AKT cascades, driving cell proliferation. GTPase-activating proteins (GAPs) subsequently inactivate KRAS by hydrolyzing GTP back to GDP. Additionally, Arf6-mediated mechanisms in KRAS regulation are depicted. The diagram emphasizes the functional significance of KRAS's activation states in regulating cellular signaling and proliferation.
Figure4. The resistance mechanism in colorectal cancer during KRAS inhibition. The model depicts the key resistance mechanisms. The figure also highlights pathway crosstalk, including the role of metabolic reprogramming and the activation of compensatory signaling nodes (e.g., YAP/TEAD and NRF2).

Round 2
Reviewer 3 Report
Comments and Suggestions for Authors
Accepted